# Fiber-Type-Specific Hypertrophy with the Use of Low-Load Blood Flow Restriction Resistance Training: A Systematic Review

**DOI:** 10.3390/jfmk8020051

**Published:** 2023-04-27

**Authors:** Brad J. Schoenfeld, Dan Ogborn, Alec Piñero, Ryan Burke, Max Coleman, Nicholas Rolnick

**Affiliations:** 1Department of Exercise Science and Recreation, CUNY Lehman College, Bronx, NY 10468, USA; 2Department of Physical Therapy, University of Manitoba, Winnipeg, MB R3T 2N2, Canada; 3Orthopedic Section, Department of Surgery, University of Manitoba, Winnipeg, MB R3T 2N2, Canada; 4Pan Am Clinic Foundation, Winnipeg, MB R3M 3E4, Canada

**Keywords:** occlusion, katsu, fiber type, muscle growth, cross-sectional area

## Abstract

Emerging evidence indicates that the use of low-load resistance training in combination with blood flow restriction (LL-BFR) can be an effective method to elicit increases in muscle size, with most research showing similar whole muscle development of the extremities compared to high-load (HL) training. It is conceivable that properties unique to LL-BFR such as greater ischemia, reperfusion, and metabolite accumulation may enhance the stress on type I fibers during training compared to the use of LLs without occlusion. Accordingly, the purpose of this paper was to systematically review the relevant literature on the fiber-type-specific response to LL-BFR and provide insights into future directions for research. A total of 11 studies met inclusion criteria. Results of the review suggest that the magnitude of type I fiber hypertrophy is at least as great, and sometimes greater, than type II hypertrophy when performing LL-BFR. This finding is in contrast to HL training, where the magnitude of type II fiber hypertrophy tends to be substantially greater than that of type I myofibers. However, limited data directly compare training with LL-BFR to nonoccluded LL or HL conditions, thus precluding the ability to draw strong inferences as to whether the absolute magnitude of type I hypertrophy is indeed greater in LL-BFR vs. traditional HL training. Moreover, it remains unclear as to whether combining LL-BFR with traditional HL training may enhance whole muscle hypertrophy via greater increases in type I myofiber cross-sectional area.

## 1. Introduction

Both type I (i.e., slow-twitch) and type II (i.e., fast-twitch) fibers can increase in size when subjected to a sufficient stimulus such as resistance training. Although evidence indicates that the hypertrophic capacity of type II fibers is substantially greater than that of type I fibers [1], it is possible that these findings may be reflective of the research designs employed in the current literature. Specifically, resistance training research generally involves the use of heavy loads (HLs) (>60% one repetition maximum (RM)), which may favor the development of type II fibers over type I fibers [2].

The seminal work of Henneman et al. [3] demonstrated that recruitment of skeletal muscle fibers follows the size principle, which states that low-threshold motor units (associated with the type I myofibers) are activated first, with higher-threshold motor units (associated with the type II myofibers) progressively recruited thereafter to meet force demands. This orderly recruitment process is thought to occur irrespective of the activity performed [4].

Based on the size principle, some researchers have surmised that HL training is required to recruit the highest threshold motor units [5]. However, this hypothesis discounts the role of fatigue in motor unit recruitment. Evidence indicates that low-load (LL) training promotes substantial type II fiber recruitment provided that sets are carried out in close proximity to muscle failure. In a study employing decomposition of electromyographic signals, Muddle et al. [6] reported that LL exercise (30% maximal voluntary contraction) performed to volitional failure resulted in the progressive recruitment of high-threshold motor units to maintain torque production, although HL tended to recruit larger motor units on average. Intriguingly, the recruitment threshold of the smallest motor units increased during LL, while thresholds for the larger motor units decreased; alternatively, HL showed no alterations in recruitment thresholds across the motor unit pool. The findings suggest that the duration of muscular activity during intense resistive exercise may result in divergent adjustments in motor unit recruitment.

Although fiber recruitment is obligatory for eliciting increases in muscle size, the duration of the stimulus may also be of importance to anabolism [7]. In this regard, LL training may induce a greater stimulus to the fatigue-resistant type I fibers given that such training involves an extended time-under-tension. Moreover, the accumulation of H+ may impede calcium binding in type II fibers, thus impelling a greater involvement of type I fibers to sustain muscular contractions [8]. It therefore has been proposed that training with low loads may preferentially hypertrophy type I myofibers, with increases in fiber cross-sectional area (fCSA) superior to that of HL training [9].

The potential to target type I fibers has important implications for sport. For example, achieving greater increases in type I fibers would further the ability of bodybuilders to maximize whole muscle hypertrophy, which is a basis upon which these athletes are judged. Moreover, athletes participating in sports that involve muscular endurance would seemingly benefit from a greater myofibrillar content in fatigue-resistant fibers. However, although some evidence does suggest a preferential hypertrophic effect on type I fibers with LL training [9], research on the topic is limited, and a recent meta-analysis of the available literature indicated similar increases in type I hypertrophy when compared to HL training [10].

Emerging evidence indicates that the use of LL training in combination with blood flow restriction (LL-BFR) can be an effective method to elicit increases in muscle size, with most research showing similar whole muscle growth of the extremities compared to HL training [11]. It is conceivable that properties unique to LL-BFR such as greater ischemia, reperfusion, and metabolite accumulation may enhance the stress on type I fibers during training compared to the use of LLs without occlusion. Accordingly, the purpose of this paper is to systematically review the relevant literature on the fiber-type-specific response to LL-BFR and provide insights into future directions for research.

## 2. Methods

### 2.1. Literature Search

The methods for this review were preregistered prior to data collection on the Open Science Framework website (https://osf.io/5eapg). To locate relevant studies, we searched PubMed/MEDLINE, Scopus, and Web of Science databases from inception to February 2023. Two researchers (RB and AP) screened the retrieved abstracts and reviewed the full texts for studies that conceivably met inclusion criteria. Inclusion required agreement between both researchers; in cases where a disagreement arose, a third researcher (B.J.S.) resolved the dispute.

The search syntax was performed using the following combination of terms: (“blood flow restriction” OR BFR OR kaatsu OR “blood flow restricted”) AND (“type I” OR “muscle fiber” OR “muscle fibre” OR “fiber-type” OR “fibre-type” OR “fiber type” OR “fibre type” OR “myofiber” OR “myofibre”). In addition, we performed secondary searches by scrutinizing the reference list of each read full text as well as examining the papers that cited the included studies via Google Scholar. The methods followed guidelines set forth by the Preferred Reporting Items for Systematic Reviews and Meta-Analyses (PRISMA) [12]. Figure 1 presents a flow chart of the search process.

### 2.2. Inclusion/Exclusion Criteria

We included studies that (1) investigated the longitudinal effects LL-BFR (defined herein as using <50% 1RM with occlusion of the proximal limb) on fiber-type-specific hypertrophy where at least one group performed resistance exercise using LL-BFR for at least 2 weeks; (2) involved combined concentric and eccentric actions; (3) included adults (18+ years of age) as participants; (4) were published in English-language peer-reviewed journals; (5) reported pre–post study changes in fiber-type-specific hypertrophy employing an objective measure of assessment.

Studies were excluded if (1) participants had pre-existing musculoskeletal disorders, cardiovascular disease, or any other condition that could be considered detrimental to resistance training performance; (2) they comprised a case report; (3) they only collected acute data; (4) they involved continuous aerobic-type exercise (defined herein as noninterrupted steady-state exercise lasting more than 15 min); (4) participants were provided with supplements intended to enhance muscle building; (5) there were insufficient numerical or graphical data to calculate relative changes.

### 2.3. Data Coding and Analysis

Data were extracted from the respective studies and coded in an Excel spreadsheet (Microsoft Corporation, Redmond, Washington) by two authors (R.B. and A.P.) using the following classifications: (1) study characteristics (author, year of publication, sample size); (2) participant demographics (age, sex, resistance training status); (3) study methods (sets, exercises, frequency, duration, %1RM, biopsy site); and (4) pre- and post-training means and standard deviations. In cases where studies lack sufficient information regarding pre–post changes, we contacted the authors to request the missing data. If we were unable to acquire data from authors, we extracted values from figures using WebPlotDigitizer online software where applicable (https://apps.automeris.io/wpd/). Any discrepancies in the extracted data were resolved through discussion and mutual consensus of the researchers. If consensus could not be reached, a third researcher (B.J.S.) resolved the dispute.

We calculated the mean percentage changes (Δ%) and effect size (ES) scores for fiber-type-specific hypertrophy for each study. Values for determining Δ% for a given outcome in a given condition were calculated as follows: ([post-training mean/pretraining mean × 100] − 100). Values for determining the ES for a given outcome in a given condition were calculated as follows: ([post-training mean − pretraining mean]/[pooled pretraining standard deviation]) [13]. When confidence intervals were provided in lieu of standard deviations, we backsolved to obtain the standard deviations using the following formula: (SD = N × [upper limit − lower limit]/3.92) [14]. We interpreted data based on estimations of the magnitude of changes between fiber types and, where applicable, between conditions.

### 2.4. Methodological Quality

As previously described [15], we assessed the methodological quality of included studies via the Downs and Black assessment tool [16], which is a 27-item checklist that addresses the following aspects of a study’s design: reporting (items 1–10), external validity (items 11–13), internal validity (items 14–26), and statistical power (item 27). Consistent with previous systematic reviews of exercise interventions, we modified the checklist by adding two items relating to participant adherence (item 28) and training supervision (item 29) [17,18,19]. Each item in the checklist is scored with a “1” if the criterion is satisfied or with “0” if the criterion is not satisfied. Based on the summary scores, studies were classified as follows: “good quality” (21–29 points); “moderate quality” (11–20 points); or “poor quality” (less than 11 points) [18,19]. Three reviewers (M.C., A.P., and R.B.) independently rated each study, and final results were settled by majority agreement.

## 3. Results and Discussion

### 3.1. Descriptive Data

A total of 11 interventions met inclusion criteria (see Table 1). Seven studies employed a parallel group design [20,21,22,23,24,25,26], two studies employed a one-group pre-/post-test design [27,28], and two studies employed a within-subject design [29,30]. Four studies solely included men [20,22,29,30] and seven studies included a combination of men and women [21,24,25,26,27,28]. Seven studies employed untrained participants [21,22,26,27,28,29,30], two employed resistance-trained participants [24,25], and two studies did not specify participant training status [20,23]. One study employed older individuals [26]; all other studies employed young participants (20–47 years old). Study duration ranged from 2 to 9 weeks. Intensities of load ranged from 20% 1RM to 50% 1RM in the LL-BFR condition. All studies solely measured fCSA of the vastus lateralis muscle.

### 3.2. Longitudinal Data

Two studies employed a one-group pre-/post-test design that investigated fiber-type-specific hypertrophy only in an LL-BFR training condition. Jakobsgaard et al. [27] submitted six healthy young untrained men and women to a six-week intervention consisting of five sets of sit-to-stand BFR exercise performed to volitional failure with 30 s inter-set rest intervals. The intervention led to an increase in type I fCSA (8.7% (ES: 0.76)) and the number of associated satellite cells and myonuclei, whereas type II fibers had no observable change (0% (ES: 0.06)) and no statistical changes in SC or myonuclei. Consistent with these findings, Bjørnsen et al. [28] submitted 13 young, untrained men and women to two five-day blocks of seven LL-BFR sessions, with each block separated by a 10-day recovery period. Participants performed four sets of knee extensions to volitional failure at 20% 1RM. Following the first block, type I and II fCSA decreased by 6% and 15%, respectively. However, significant hypertrophy was observed 10 days after the second block, with greater increases observed in type I compared to type II fibers (19% (ES: 0.64) vs. 11% (ES: 0.51), respectively). In a follow-up study to their prior investigation [28], Bjornsen et al. [30] employed a within-subject design using the same basic protocol (two 5-day blocks of seven LL-BFR sessions separated by a 10-day rest period) to determine whether their previously observed results were dependent on training to muscle failure. Seventeen young, untrained men performed four sets of LL-BFR (20% 1RM) knee extensions with one leg assigned to failure and the other not to failure (30-15-15-15 repetitions). Intriguingly, type I fibers atrophied by ~10% in the leg that trained to failure; no fCSA changes were observed for type I fibers in the nonfailure leg. Myofiber area of type II fibers did not meaningfully change from baseline to post-study in either condition.

One study compared LL-BFR training to a nontraining control using a randomized, parallel group design [26]. The LL-BFR training consisted of four sets of leg extension exercise at 30% 1RM training with all sets carried out to volitional failure; training sessions were performed 3 days/week. Results showed that LL-BFR elicited similar increases in type I and type II fCSA (18.1% (ES: 0.62) vs. 22.2% (ES: 0.53), respectively).

Three studies directly compared fiber-type-specific hypertrophy between LL and LL-BFR. Yasuda et al. [20] allocated five young men to perform either LL-BFR (n = 3) at 20% 1RM or LL exercise without occlusion at the same intensity of load for 2 weeks. Training consisted of two daily sessions of the squat and leg curl. Both groups performed three sets of 15 repetitions for each exercise. Results showed greater hypertrophy in type II compared to type I myofibers (27.6% (ES: 0.77) vs. 5.9% (ES: 0.16), respectively) in the LL-BFR group; no appreciable changes were seen in either fiber type for the LL group. Nielsen et al. [22] allocated 18 young untrained men to perform four sets of knee extension exercise at 20% 1RM either using BFR or without occlusion. The LL-BFR group performed sets to concentric failure while the nonoccluded training was work-matched to that of the LL-BFR group. Both groups took part in 23 training sessions carried out over 19 days. Results showed that LL-BFR elicited increases in type I and type II fCSA, which were of similar magnitude (35% (ES: 1.87) and 37% (ES: 1.99), respectively); the LL condition showed no appreciable pre-/post-study changes in either outcome. Pignanelli et al. [29] randomly assigned the legs of 10 young men to perform three sets of single-leg Smith machine squats at 30% 1RM with LL-BFR and LL without occlusion; both conditions trained to momentary muscular failure on each set. Training was carried out 3 days per week for 6 weeks. Results showed that LL-BFR similarly increased CSA of both fiber types. Intriguingly, LL exercise without occlusion also showed appreciable increases in fCSA, with hypertrophic differences modestly favoring type I compared to type II fibers (15.8% (ES: 0.56) vs. 9.0% (ES: 0.38), respectively).

Two studies compared HL training to a program involving a combination of LL-BFR and HL training. Bjørnsen et al. [25] randomized 17 nationally ranked powerlifters to a group that performed either front squats at 60–85% 1RM or a group that interspersed two blocks (during weeks 1 and 3) of LL-BFR (~30% of 1RM) training into this 6.5-week protocol. The LL-BFR consisted of five weekly sessions of four sets of front squats, with the first and last sets performed to volitional failure. Results indicated preferential increases in type I fCSA for the group that integrated LL-BFR compared to the traditional training protocols (12% (ES: 0.80) vs. 0% (ES: 0.00), respectively) and myonuclear number (18% vs. 0%, respectively). No changes were observed in Type II fCSA for either group. Ultrasound-derived measures of vastus lateralis CSA favored the LL-BFR group compared to the traditional group (7.7% vs. 0.5%, respectively); these findings showed a strong correlation with the CSA increases observed in type I fibers (*r* = 0.81). More recently, Hansen et al. [23] randomized 18 active young men and women to perform a 6-week program consisting of either conventional resistance training (2–4 sets at 70–90% 1RM) or conventional resistance training combined with LL-BFR (four sets at 20% 1RM) on the leg press and leg extension exercises. The LL-BFR group alternated performing sessions of only LL-BFR (weeks 1, 3, and 5) and a combination of LL-BFR and HL training (weeks 2, 4, and 6). In contrast to the findings of Bjornsen et al. [25], only the HL group appreciably increased type I fCSA, whereas type II fCSA increased similarly between conditions.

Two studies investigated the fiber-type-specific response between LL-BFR and traditional HL training. Davids et al. [24] randomized resistance trained subjects to perform a variety of lower body exercises using either HL (75–80% 1RM) or LL-BFR (30–40% 1RM). Training was carried out 3 days a week for 9 weeks. Results showed that LL-BFR elicited greater fCSA increases in type I myofibers compared to HL training (10.6% (ES: 0.32) vs. 1.7% (ES: 0.07), respectively); hypertrophy of type II myofibers modestly favored HL compared to LL-BFR (17.7% (ES: 0.53) vs. 12.6% (ES: 0.36), respectively). Sieljacks et al. [21] randomly assigned untrained men and women to a thrice weekly program comprising four sets of knee extension using either HL (70% 1RM) or LL-BFR (~30% 1RM). The HL group performed 12 repetitions per set with 3 min rest intervals, whereas LL-BFR performed sets to volitional failure with 30 s rest intervals. Results indicated no appreciable change in type I or II fCSA for either condition.

### 3.3. Methodological Quality

Qualitative assessment of the studies via the Downs and Black checklist indicated a median score of 22 (range: 13 to 24 points). Nine studies were deemed to be of good quality [21,23,24,25,26,27,28,29,30], two studies were classified as being of moderate quality [20,31], and no studies were found to be of poor quality.

### 3.4. Reconciling Current Data

Researchers have speculated that LL-BFR may provide a greater hypertrophic stimulus to type I myofibers compared to traditional HL training [8,32]. Indeed, acute research assessing the heat shock protein (HSP) response to resistance exercise with LL-BFR provides a hypothetical basis for preferential type I hypertrophy. Various HSPs, including HSP70 and αB-crystallin, have been shown to facilitate repair of proteins subjected to exercise-induced stress (e.g., muscle damage, ischemia, etc.) [33]. Pursuant to such stress, these HSPs translocate from their unbound state in the cytosol and accumulate in the perturbed myofibers to initiate remodeling. Theoretically, fibers that show a greater HSP infiltration would have achieved a greater level of stimulation that may, in turn, drive long-term muscular adaptations. To this end, Cumming et al. [34] demonstrated that HSP70 levels increased over 48 h after an acute bout of LL-BFR training, with greater observed responses in type I fibers compared to type II fibers. Greater glycogen depletion was observed in type I fibers 48 h post-exercise, which correlated with the elevated HSP70 response in these fibers. Similarly, Bjørnsen et al. [35] found that translocation of αB-crystallin and HSP70 to the cytoskeleton was greater in type I than in type II fibers following high-volume blocks (5 days) of LL-BFR. Consistent with the findings of Cumming et al. [34], glycogen depletion was greater in type I vs. type II myofibers. Taken together, these findings suggest that the stress response to BFR is more pronounced in type I compared to type II myofibers, which conceivably may result in greater hypertrophy of slow-twitch fibers.

While the acute data provide a logical rationale for preferential type I hypertrophy with LL-BFR training, results of longitudinal research on the topic reviewed herein are somewhat equivocal. Studies that employed a single-group pre–post study design or generally showed that LL-BFR elicited greater type I vs. type II hypertrophy [27,28]. Alternatively, a follow-up study that used a within-subject design to tease out the potential confounding effects of training to failure during LL-BFR did not detect a preferential hypertrophic effect on type I fibers [30]. However, results may have been influenced by an interaction between the training status of the sample and training protocol. Specifically, the authors noted that the participants were largely sedentary and thus speculated that the intense, high-volume protocol (two blocks of seven LL-BFR sessions carried out in 5 days) may have led to heightened muscle damage in the type I fibers that continued to undergo remodeling at the time of final biopsy. This hypothesis was supported by elevated muscle echogenicity assessed by ultrasound imaging, which is indicative of elevated tissue disruption. Although speculative, the authors therefore hypothesized that muscle fibers may ultimately have shown a delayed hypertrophic response that would have manifested had follow up biopsies been taken at later time points. A more recent study that compared LL-BFR training to a nontraining control group showed robust increases in the size of both type I and type II myofibers, with negligible differences between fiber types. Taken as a whole, these findings suggest that although research generally shows similar increases in whole muscle growth between LL-BFR and HL training [11], the results may arise from differential hypertrophy between fiber types, with LL-BFR inducing greater type I hypertrophy and HL greater type II hypertrophy. This hypothesis remains speculative since none of the aforementioned studies included an HL group for direct comparison.

It has been theorized that nonoccluded LL training may target type I myofiber hypertrophy. Studies directly comparing LL-BFR to LL training without occlusion provide mixed results in regard to preferential increases in type I fCSA. Yasuda et al. [20] showed that LL-BFR elicited substantially greater type II vs. type I hypertrophy (27.6% vs. 5.9%, respectively) while a cohort performing LL training without occlusion showed negligible hypertrophy in either fiber type. However, the very small sample (n = 5) makes it difficult to draw practical inferences from the data. Moreover, the proximity to failure in the LL-BFR group is unclear, raising questions as to whether the type I fibers were sufficiently stimulated to induce hypertrophy. Conversely, both Nielsen et al. [22] and Pignanelli et al. [29] found similar LL-BFR-induced hypertrophy between type I and type II myofibers. However, Nielsen et al. [22] showed negligible pre–post hypertrophic changes in the nonoccluded LL condition while Pignanelli et al. [29] reported slightly greater type I fSCA increases in LL without occlusion compared to LL-BFR. The discrepancies may be explained by the fact that Nielsen et al. [22] employed a work-matched design where the nonoccluded LL condition seemingly terminated sets far short of failure whereas Pignanelli et al. [29] had both groups train to volitional failure. Overall, the limited current data preclude the ability to determine if LL-BFR indeed promotes an added stimulus to type I fibers over and above that of LL training without occlusion when sets are performed with a high level of effort.

Studies directly comparing fiber-type-specific hypertrophy between LL-BFR and traditional HL training are contradictory. Davids et al. [24] found greater type I hypertrophy favoring LL-BFR compared to HL training, while Sieljacks et al. [21] reported no between-group differences. Curiously, changes in fCSA were negligible in both fiber types across conditions in Sieljacks et al. [21], which is inconsistent with the prevailing body of literature. The authors surmised that the low-volume, single-exercise protocol may have been insufficient to promote appreciable hypertrophy [21], although other research reports substantial fCSA increases when employing similar protocols in a comparable population [29]. An alternative explanation for the discrepant findings remains elusive.

If LL-BFR does indeed promote preferential type I hypertrophy, then conceivably it could be beneficial to integrate the strategy into conventional RT programming. However, current evidence is conflicting on the topic. Bjornsen et al. [25] found an appreciably greater increase in type I fibers when training with LL-BFR vs. HL, whereas Hansen et al. [23] found greater type I hypertrophy with HL vs. LL-BFR. Discrepancies between studies may be related to the training status of the samples. Specifically, participants in Bjornsen et al. [25] were elite powerlifters while Hansen et al. [23] classified participants as “active” and, although training status was not mentioned, they seemingly lacked consistent experience with resistance exercise. Although it is tempting to speculate that the hypertrophic benefits of combined LL-BFR and HL may be specific to well-trained individuals, the fact that other studies show that LL-BFR promotes substantial type I hypertrophy in untrained participants [22,27,28,29] raises doubts regarding this hypothesis.

### 3.5. Limitations and Future Directions

Although LL-BFR represents an intriguing strategy to preferentially target hypertrophy of type I myofibers, limitations of the current research preclude the ability to draw strong inferences on the topic. First, the designs of studies directly comparing type I fiber hypertrophy in LL-BFR are heterogenous. As noted above, some studies investigated the fiber-type-specific effects of LL-BFR in isolation, others compared LL-BFR to LL or HL training without occlusion, and yet others assessed the combination of LL-BFR and HL training. While the sum total of results from these studies provides a basis for triangulation of data, additional research is needed to directly compare fCSA adaptations in LL-BFR vs. HL training protocols. Moreover, the inherent heterogeneity of study designs precluded our ability to carry out a meta-analysis of findings and, consequently, to quantify the magnitude of effects on outcomes.

Second, all studies on the topic to date have assessed fiber type adaptations in the vastus lateralis. Results therefore cannot necessarily be extrapolated to other skeletal muscles. For example, given differences in the use of the upper body vs. lower body muscles in activities of daily living, it is possible that myofibers in these bodily regions may respond differentially to LL-BFR. Moreover, the fiber type composition of a given muscle might play a role in the adaptative response to LL-BFR, whereby muscles with higher percentages of a given fiber type could show divergent results (the vastus lateralis generally comprises a mixture of fiber types). These hypotheses warrant further exploration.

Third, commonly used methods to assess fiber-type-specific hypertrophy have questionable accuracy and can be influenced by various factors including the depth of biopsy sampling, fiber count of the acquired sample, and number of obtained biopsies [36]. Horwath et al. [36] reported large coefficients of variation between vastus lateralis biopsy samples in type I and type II fibers (13% and 14%, respectively), with results deviating by as much as 42% to 48% within subjects. This raises skepticism regarding the confidence that can be placed on the current data when attempting to draw evidence-based inferences.

Fourth, studies to date have used relatively short durations, some lasting as few as 2 weeks, with a maximum length of 9 weeks. This raises the question as to whether the short-term beneficial effects achieved in some studies would manifest across longer time periods. Moreover, might LL-BFR be a strategy that can be intermittently implemented over the course of a training mesocycle to enhance muscular adaptations? These questions warrant further investigation.

Finally, current research has almost exclusively involved younger participants. Research indicates that the age-related loss of muscle tends to be most specific to type II fibers [37]. Given the greater percentage of type I fibers in older muscle, it is conceivable that LL-BFR might promote greater overall hypertrophy with advancing age. Indeed, the only controlled study to date on the topic showed similar, robust hypertrophy of both type I and type II fibers following 6 weeks of LL-BFR training in a cohort of older individuals [26], whereas evidence indicates that traditional heavier load training preferentially hypertrophies type II myofibers with minimal changes in type I fiber size [38]. Future research should further explore if and how LL-BFR might differentially affect older vs. younger individuals.

## 4. Conclusions

The current body of literature suggests that the magnitude of type I fiber hypertrophy is at least as great as, and sometimes greater than, type II hypertrophy when performing LL-BFR. This finding is in contrast to HL training, where the magnitude of type II fiber hypertrophy tends to be substantially greater than that of type I myofibers [1]. The effects of LL-BFR on type I fCSA appear to be more pronounced with very frequent training bouts (>5 days/week) [22,25,28], although some studies show a preferential effect on type I fibers with lower training frequencies [24,27]. The role of training frequency in fiber-type-specific hypertrophy induced by LL-BFR warrants additional study.

Despite the intriguing preliminary evidence for preferential type I fiber development, limited data directly compare training with LL-BFR to nonoccluded LL or HL, thus precluding the ability to draw strong inferences regarding whether the absolute magnitude of type I hypertrophy is indeed greater in LL-BFR vs. traditional HL training. Moreover, it remains unclear as to whether combining LL-BFR with traditional HL training may enhance whole muscle hypertrophy via greater increases in type I fCSA. Further research is needed to improve our understanding of the topic and its potential practical implications for exercise program design.

## Figures and Tables

**Figure 1 jfmk-08-00051-f001:**
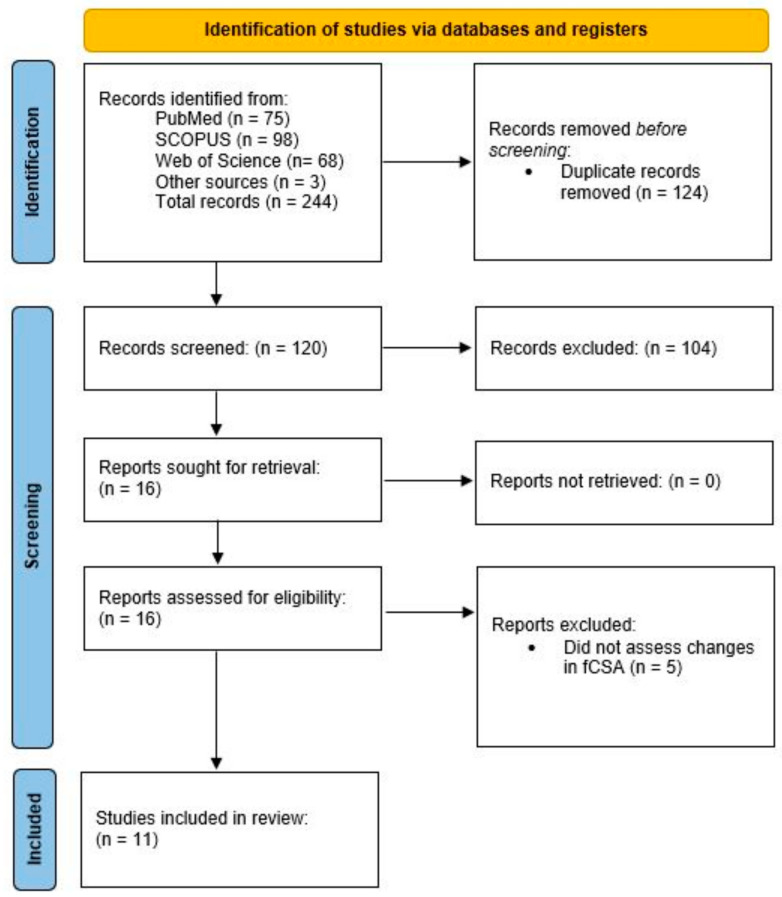
Flow chart of search process.

**Table 1 jfmk-08-00051-t001:** Summary of longitudinal studies investigating the effects of LL-BFR on fiber-type-specific hypertrophy.

Study	Sample	Design	Duration	Findings
Bjørnsen, Wernbom, Kirketeig et al. [25]	17 nationally ranked powerlifters	Parallel design. Participants randomly assigned to either an HL group who performed front squats at 60–85% 1RM or an LL-BFR group who integrated 2 blocks (weeks 1 and 3) of 5 LL-BFR (~30% 1RM at a pressure of 120 mmHg) front squat sessions into the traditional training. Traditional training consisted of 6–7 sets per session whereas LL-BFR training consisted of 4 sets (first and last set to voluntary failure) with 30 s rest intervals; both groups trained 5 days/week.	6.5 weeks	LL-BFR elicited a greater increase in type I fCSA compared to HL training (12% (ES: 0.80) vs. 0% (ES: 0.00), respectively); type II fCSA remained relatively unchanged over the study period.
Bjørnsen, Wernbom, Løvstad et al. [28]	13 young, untrained men and women	One group, pre-/post-test design. Participants performed two 5-day blocks of 7 LL-BFR (20% 1RM at a pressure of 90–100 mmHg) sessions, separated by a 10-day rest period. Exercise consisted of 4 sets of unilateral knee extensions to volitional failure with 30 s rest intervals.	3 weeks	Type I and type II fCSA decreased after the first training block (−6% and −15%, respectively). Alternatively, type I fCSA increased to a greater extent than type II fCSA after the second training block (19% (ES: 0.64) vs. 11% (ES: 0.51), respectively).
Bjørnsen, Wernbom, Paulsen, Berntsen et al. [30]	17 young, untrained men	Within-subject design. Participants had their legs randomized to perform two 5-day blocks of 7 LL-BFR (20% 1RM sessions at a pressure of 90–100 mmHg), separated by a 10-day rest period, either to volitional failure or not to failure (30-15-15-15 repetitions). Exercise consisted of 4 sets of unilateral knee extensions with 30 s rest intervals.	3 weeks	Type I fCSA decreased by 10.3% (ES: 0.70) after 10-days post-training in the failure leg while no appreciable change was observed in the nonfailure leg. Type II fCSA did not appreciably change from baseline in either condition.
Davids et al. [24]	21 young, trained men and women	Parallel group design. Random assignment to either HL training (8 repetitions at 10 RM) or LL-BFR (~30% to 50% 1RM at 60% arterial occlusion pressure). Exercise consisted of combinations of the barbell back squat, leg press, Bulgarian split squat, and leg extension. Both groups performed 4 sets of 2–3 exercises 3 days/week. Intensity of effort was standardized so that both conditions trained within 1-4 repetitions from failure.	9 weeks	Type I fCSA increased to a greater extent in LL-BFR compared to HL (10.6% (ES: 0.32) vs. 1.7% (ES: 0.07), respectively); type II hypertrophy favored HL vs. LL-BFR (17.7% (ES: 0.53) vs. 12.6% (ES: 0.36), respectively).
Hansen et al. [23]	18 young men and women	Parallel group design. Random assignment to perform either HL training (70–90% 1RM) or an LL-BFR group who alternated weekly between LL-BFR (20% 1RM at a pressure of 110 mmHg) and HL Exercise consisted of the leg press and leg extension carried out 4 days/week; HL performed 2–4 sets per exercise of 3–10 repetitions with 2 min rest intervals whereas LL-BFR performed 4 sets to volitional failure with 30 s rest intervals	6 weeks	Type I fCSA increased to a greater extent in the HL group compared to LL-BFR (12% (ES: 0.81) vs. 2.5% (ES: 0.17), respectively); type II fCSA increased similarly between groups (~16%).
Jakobsgaard et al. [27]	6 healthy young untrained men and women	One group, pre-/post-test design. Subjects performed 5 sets of sit-to-stand LL-BFR (pressure of 100–180 mmHg) to volitional failure with 30 s rest intervals carried out 3 days/week.	6 weeks	Type I fCSA increased to a greater extent than type II fCSA (8.7% (ES: 0.76) vs. 0% (ES: 0.06), respectively).
Nielsen et al. [22]	18 young untrained men	Parallel group design. Participants assigned to LL training (20% 1RM) either using BFR (at a pressure of 100 mmHg) or performing exercise without occlusion. Training consisted of 4 sets of knee extension exercise carried out once or twice daily for a total of 23 sessions performed within 19 days). LL-BFR carried out sets to volitional failure, whereas LL carried out sets in a work-matched fashion.	3 weeks	For LL-BFR, type I and type II fCSA increased similarly from baseline to 3 days post-study (35% (ES: 1.87) and 37% (ES: 1.99), respectively). Alternatively, type I and type II fCSA did not appreciably change from pre–post study in LL.
Pignanelli et al. [29]	10 young, untrained men	Within-subject design. Participants had their legs randomly assigned to LL training (30% 1RM) either using BFR (at 60–70% of the lowest effective occlusive pressure) or performing exercise without occlusion. Training consisted of 4 sets of single leg squats to volitional failure with 100 s rest intervals carried out 3 days/week.	6 weeks	Type I fCSA increased to a greater extent in LL compared to LL-BFR (15.8% (ES: 0.56) vs. 10.1% (ES: 0.38), respectively); type II fCSA increased similarly between conditions (~9% (~ES: 0.30)).
Sieljacks et al. [21]	34 young, untrained men and women	Parallel group design. Random assignment to either HL training (70% 1RM) or LL-BFR (~30%/1RM at a pressure of 97 mmHg) training carried out to volitional failure. Training consisted of 4 sets of knee extension exercise carried out 3 days/week. HL performed 12 repetitions with 3 min rest intervals whereas LL-BFR performed sets to volitional failure with 30 s rest intervals	6 weeks	Minimal pre–post study changes were observed in LL-BFR and HL conditions for both type I fCSA (−2.4% (ES: −0.20) and −2.3% (ES: −0.11), respectively) and type II fCSA (2.6% (ES: 0.15) and −2.3% (ES: −0.11), respectively).
Wang et al. [26]	23 older, untrained men and women	Parallel group design. Random assignment to either LL-BFR (~30%/1RM at a pressure of 97 mmHg) training carried out to volitional failure or a nontraining control. LL-BFR training consisted of 4 sets of knee extension exercise with 30 s rest intervals carried out 3 days/week.	6 weeks	Type I and type II fCSA increased to a similar extent (18.1% (ES: 0.62) vs. 22.2% (ES: 0.53), respectively).
Yasuda et al. [20]	5 young men	Parallel group design. Participants performed either LL-BFR (20% 1RM at a pressure of 160–240 mmHg) training or LL training without occlusion. Exercise consisted of 3 sets of 15 repetitions of the squat and leg curl with 30 s rest intervals carried out twice daily for the duration of the study period.	2 weeks	For LL-BFR, type II fCSA increased to a greater extent than type I fCSA (27.6% (ES: 0.77) vs. 5.9% (ES: 0.16), respectively). Type I and type II fCSA showed minimal pre–post study changes in LL (−2.1% (ES: −0.11) and 0.5% (ES: 0.7), respectively).

Abbreviations: LL: low load; LL-BFR: low-load blood flow restriction; fCSA: HL: high load; fiber cross-sectional area; 1RM: 1 repetition maximum; mmHg: millimeters of mercury; ES: effect size.

## Data Availability

No new data were created in this study. Data sharing is not applicable to this article.

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
