# Peer review of "Fiber-Type-Specific Hypertrophy with the Use of Low-Load Blood Flow Restriction Resistance Training: A Systematic Review"

_jfmk, 2023, doi:10.3390/jfmk8020051_

Round 1
Reviewer 1 Report
This paper is a well written piece and is in a coherent structure and has been able to address all the main ideas that a systematic review requires. I have a few comments.
Figure 1: in some fields the text is a bit cut off and not well discernible.
Lines 188-189: Four studies included only men while 6 included a combination of men and women. This makes a total of 10. What did the 1 remaining study include? Kindly state this in addition.
Lines 201: “increase in satellite cell and myonuclei” – increase in what? Numerical density?
Line 236: - Pax7+ cells are satellite cells? Please use same nomenclature throughout the paper, since it may confuse readers.
Did any study separately study type 2a and 2x/d fibres? What about hybrid fibres?
Please indicate which muscle was studied in each case.
Author Response
Please see attachement

Reviewer 2 Report
The manuscript covers the interesting topic reggarding the hypertrophic responses to low-load vs high load resistive exercise training. The authors focuse on the pecularities of the blood-flow restriction training regime combined with the LL resistive training. The authors collected data of 11 experimental papers dedicated to the effects theswe training regimes. The authors performed the great work analyzing the effects of LL-BFR training as compared to the LL-regime effects on the slow- and fast twitch fiber cross-sectional areas. They concluded that LL and presumably LL+BFR training led to the predominant hypertrophy of the slow-twitch fibers, although this concusion was not supported by the majority of the analyzed experimental reports. The analysis is useful but to my mind the review lacks many important items. And this lack makes the presented manuscript nebulous and difficult to read for the scientific readers.
Here are my concerns:
- it is necesary to compare effects of 3 states HL, LL and LL+BFR and reveal the specific effects of each regime. It is difficult to understand what the BFR really does.
- it is necessary to make at least brief description of the possible mechanisms of the hypertrophy development in each of the analyzed training modes. The authors tell us nothing about pecularitie of the energy metabolism control (e.g. AMPK, PGC1alpha etc) and anabolic regulation (mTORc1, ribosome biogenesis etc).
- it is strange that the authors pay much attention to the expresion of HSPm since HSPs are not the direct participants of the muscle mass control.
The paper should be rewritten and returned to the reviewers again.
Reviewer 3 Report
L23. I suggest to change “traditional methods.” to “traditional HL training”.
L46. I suggest to clarify “carried out with a high intensity of effort.”
Reference 8 seems to be a non-peer reviewed source. I suggest to reconsider or clarify.
L83. Font size of webadress needs change.
Fig 1. Please ensure that all can be read in Figure 1. Some information is clipped.
Fig 1 indicates 11 studies but L20 in abstract only 10. Please clarify.
L163. Please define “.d%”. Is this delta change (%).
L186. Consult author guidelines for providing “(20) (21) (22) (23) (24) (25) (26)”. This should be “[2-26]”. Revise throughout the manuscript.
L202. I suggest to change “minimal growth (0%” to “no growth (0%”. In addition, I suggest to consider hypertrophy in stead of growth is applicable to development from young to an older age for example. Please reconsider to change “growth” to “hypertrophy” throughout the manuscript.
L382. I suggest to add the fibre type composition of the m.vastus lateralis.
Please consult author guidelines for the format of the references.
Round 2
Reviewer 2 Report
I have read the authors' response, and understood that the authors focused on the descriptive narration of the effects of LL-BFR on the slow and fast fiber hypertrophy. From this point of view the review paper is without any significant problems. Maybe this paper is not so interesting for me and most molecular physiologists as if it include the mechanistic discussion.